# Production and application of manure nitrogen and phosphorus in the United States since 1860

Zihao Bian[1], Hanqin Tian[1], Qichun Yang[2], Rongting Xu[1], Shufen Pan[1], Bowen Zhang[3]

[1]International Center for Climate and Global Change Research, School of Forestry and Wildlife Sciences, Auburn University, Auburn, AL 36849, USA

[2]Department of Infrastructure Engineering, The University of Melbourne, Parkville 3010, Australia

[3]Department of Environment, Geology, and Natural Resources, Ball State University, Muncie, IN 47306 USA

*Correspondence to*: Hanqin Tian (tianhan@auburn.edu)

**Abstract:** Livestock manure nitrogen (N) and phosphorus (P) play an important role in
biogeochemical cycling. Accurate estimation of manure nutrient is important for assessing
regional nutrient balance, greenhouse gas emission, and water environmental risk. Currently,
spatially explicit manure nutrient datasets over century-long period are scarce in the United
States (U.S.). Here, we developed four datasets of annual animal manure N and P production and
application in the contiguous U.S. at a 30 arc-second resolution over the period of 1860-2017.
The dataset combined multiple data sources including county-level inventory data, as well as
high-resolution livestock and crop maps. The total production of manure N and P increased from
1.4 Tg N yr$^{-1}$ and 0.3 Tg P yr$^{-1}$ in 1860 to 7.4 Tg N yr$^{-1}$ and 2.3 Tg P yr$^{-1}$ in 2017, respectively.
The increasing manure nutrient production was associated with increased livestock numbers
before the 1980s and enhanced livestock weights after the 1980s. The manure application
amount was primarily dominated by production and its spatial pattern was impacted by the
nutrient demand of crops. The intense-application region mainly enlarged from the Midwest
toward the Southern U.S., and became more concentrated in numerous hot spots after the 1980s.
The South Atlantic-Gulf and Mid-Atlantic basins were exposed to high environmental risks due
to the enrichment of manure nutrient production and application from the 1970s to the period of
2000-2017. Our long-term manure N and P datasets provide detailed information for national and
regional assessments of nutrient budgets. Additionally, the datasets can serve as the input data for
ecosystem and hydrological models to examine biogeochemical cycles in terrestrial and aquatic
ecosystems. Datasets are available at https://doi.org/10.1594/PANGAEA.919937 (Bian et al.,

46    2020).

**Keywords:** Manure; Nutrient; Nitrogen; Phosphorus; Production; Application

# 1 Introduction

Animal manure, as a fertility package, is a traditional source of nutrients and can provide abundant nitrogen (N), phosphorus (P), and potassium for cropland and pasture. Animal manure nutrients circulate widely in the Soil-Plant-Animal system and are highly involved in global nutrient cycling (Bouwman et al., 2013; Sheldrick et al., 2003). Although synthetic fertilizer has been widely used since the mid-20th century, livestock excreta is still the major nutrient source in agricultural soils, accounting for approximately 18% and 28% of the total N and P inputs to global cropland, respectively (Sheldrick et al., 2003; Zhang et al., 2020). Moreover, the total global animal manure N and P production has exceeded global fertilizer use (Bouwman et al., 2009). Therefore, the efficient recycling of manure can potentially meet the growing nutrient demand of crops. The circular nutrient source provided by manure enables nations to sustain their agricultural production with less reliance on imported fertilizer, especially mineral P fertilizer (Koppelaar and Weikard, 2013; Powers et al., 2019). Different from N which can be fixed from the atmosphere through microbial symbiosis with plants and the Haber-Bosch process, P is a rock-derived nutrient and there is no biological or atmospheric source for P. The limited and unevenly distributed P-rich rocks can threaten food security and have raised concerns in many resource-limited countries, including the United States (U.S.) (Amundson et al., 2015). Enhanced recovery nutrients from manure can not only increase agricultural dependence, but may also reduce nutrient losses out of the Soil-Plant-Animal system. Additionally, the improvement of livestock operations in recent decades also facilitated the recoverability and utilization of animal manure (Kellogg et al., 2000).

Although the application of manure and fertilizer enhanced crop production, excessive nutrients might leave the Soil-Plant-Animal system through the biogeochemical flow and potentially

contaminate the environment if not properly managed (Mueller and Lassaletta, 2020; Zanon et
al., 2019).  Specifically, agricultural land is a sink for anthropogenic N and P inputs (e.g.
synthetic fertilizer, manure, atmospheric deposition), and simultaneously acts as N and P sources
for aquatic systems as well as a N source for atmosphere (Bouwman et al., 2013; Elser and
Bennett, 2011; Schlesinger and Bernhardt, 2013). The major N gaseous loss from fertilizer use
and animal excreta includes the emissions of ammonia ($NH_3$), nitrous oxide ($N_2O$), and nitric
oxide. $NH_3$ can react with other air pollutants and form aerosols to reduce visibility and threaten
human health (Bouwman et al., 2002; Xu et al., 2018), and $N_2O$ is one of the most important
greenhouse gasses (Davidson, 2009). $N_2O$ emission from animal manure is one of the major
contributors to global anthropogenic $N_2O$ emissions (Tian et al., 2020). Additionally, large
fractions of the N and P applied to cropland lost through leaching, erosion, and surface runoff
and are transported into rivers toward lakes and coastal oceans (Smith et al., 1998; Van Drecht et
al., 2005). Excess N and P could dramatically impair freshwater and coastal ecosystems, causing
eutrophication, hypoxia, and fish-killing (Garnier et al., 2015; Smith et al., 2007). Oxygen-
depleted marine coastal "dead zones" associated with nutrient-stimulated algal blooms continue
to expand. For example, the northern Gulf of Mexico is one of the largest dead zones in the
world and the hypoxic area often exceeds 15,600 $km^2$ in midsummer (1968-2016) (Del Giudice
et al., 2019).
Considering the importance of manure nutrient on crop production, greenhouse gas emission,
and water pollution, it is vital to have a better understanding of livestock manure nutrient
production and application at national or even global scales (Potter et al., 2010; Sheldrick et al.,
2002; Tian et al., 2016). Quantized and spatialized manure nutrient data can help stakeholders
find a local recyclable nutrient source or make strategies to minimize N and P losses. Currently,
most studies only provided county-level manure nutrient production data in the U.S., with short
periods (Kellogg et al., 2000; Ruddy et al., 2006). Nevertheless, terrestrial biosphere models
usually require spatially explicit manure nutrient input data to simulate the anthropogenic effect
on biogeochemical cycles since the preindustrial period (Tian et al., 2019). Studies focusing on
soil nutrient storage change and legacy soil nutrient also need long-time series manure nutrient
data (MacDonald et al., 2012; Rowe et al., 2016). Moreover, previous studies usually assumed
that nutrient excretion per animal is constant over time when quantifying nutrient production
based on livestock number, which may lead to uncertainties (Zhang et al., 2020). Geographically
explicit manure nutrient application in cropland (excluding pasture), as the direct nutrient input
for the soil-crop system, hasn't been specifically estimated across the U.S. In this study, our
objectives are to (1) develop grid-level manure N and P production datasets in the U.S. based on
county-level livestock populations, dynamic livestock weight over time, and high-resolution
livestock distribution maps; (2) develop grid-level manure N and P application in cropland
datasets by integrating manure nutrient production and nutrient demand of crops; (3) investigate
the spatiotemporal patterns of manure nutrient production and application based on these
datasets, and (4) further identify regions with a high risk of excessive nutrient loading. The four
datasets display the masses of manure N and P per area in each 30×30 arc-second grid-cell
during 1860-2017. The datasets can be used to drive ecosystem, land surface, and hydrological
models to simulate manure-induced greenhouse gas emissions and nutrient loadings.

## 2 Methods

Datasets of manure N and P production and application were developed by incorporating
multiple datasets (Table 1). The geographically explicit manure N and P production data were
first calculated based on county-level livestock populations, dynamic livestock weights, and
livestock distribution maps. Then the crop nutrient demand maps were developed by merging
cropland distribution maps with crop-specific harvest area and nutrient assimilative capacities.
Finally, the spatially explicit manure N and P application data were estimated by incorporating
county-level manure production, recoverability factors, cropland fraction, and cropland nutrient
demand maps. To facilitate studying the impact of manure nutrients on water quality, we further
analyzed the average annual manure production and application in four decades (the 1860s,
1930s, 1970s, and 2010-2017) across the major 18 basins (Fig 1).
**2.1 Manure nutrient production**
Manure nutrient production refers to the animal excretion in this study. The county-level manure
N and P production during 1930-2017 were calculated based on the livestock population, animal
body weight, and nutrient excretion rates according to the method (Eq. 1) proposed by Puckett et
al. (1998).
County-level manure nutrient production was calculated as follows:
$$Pro_{x,c} = \sum_{i=1}^{n} Pop_{i,c} \cdot W_i \cdot Er_{x,i} \cdot Days \qquad (1)$$

where $Pro_{x,c}$ is the annual manure nutrient $x$ (N or P) production in county $c$ (kg N/P yr$^{-1}$); $i$ is
animal type; $Pop_{i,c}$ is the county-level animal population (head); $W_i$ is the annual average live
body weights of animal (kg); $Er_{x,i}$ represents the excreted manure nutrients rate per unit weight
of animal (kg N/P kg$^{-1}$ day$^{-1}$) (Table S1); $Days$ is the number of days in the life cycle of animal
within a year.
Data of livestock and poultry population were derived from the U.S. Department of Agriculture
(USDA) census reports from 1930 to 2017 at 4- or 5-year intervals. Eleven livestock and poultry
categories were considered in this study, including beef cows, milk cows, heifers, steers, hogs,
sheep, horses, chickens, pullets, broilers, and turkeys. Livestock population data for the recent
five census reports (1997–2017) can be directly collected from the USDA Census Data Query
Tool. Livestock population data before 1997 were collected from Cornell Institute for Social and
Economic Research Data Archive (1949–1992), or manually digitalized from the USDA reports
(1930–1945). More details of data collection, methods of dealing with missing data can be found
in Yang et al. (2016). Annual average live weights of livestock and poultry, including cattle,
hogs, sheep, broilers, chickens, and turkeys, were derived from the USDA Economic Research
Service. We developed annual manure nutrient production by assuming a linear change between
every two census years.
Global Livestock Impact Mapping System (GLIMS) provided gridded livestock population
maps at a resolution of 30 arc-second (https://livestock.geo-wiki.org/home-2/). These maps were
developed according to statistical relationships between livestock inventory data and multiple
environmental variables, including climate, land cover, and human activities (Robinson et al.,
2014). Combining with the GLIMS data, we spatially allocated manure nutrient production
within each county. The grid-level manure nutrient production was first calculated based on the
GLIMS data, and the total quantity of manure nutrient production in each county was obtained
by calculating the sum of productions in all grid-cells within each county. Then, we calculated
ratios of USDA-based county-level manure nutrient production to GIMS-based county-level
data, and these ratios were used to adjust grid-cell values within each county. After this step, the
developed grid-level products were in line with USDA-based annual county-level data in total
quantities, with the spatial pattern inside each county inherited from the GLIMS-based manure
nutrient production data (Eq. 2).
$$Pro_{x,j} = GPro_{x,j} \cdot \frac{Pro_{x,c}}{GPro_{x,c}} \qquad (2)$$
where $Pro_{x,j}$ is manure nutrient production in grid-cell $j$ (kg N/P km$^{-2}$ yr$^{-1}$); $GPro_{x,j}$ is manure
nutrient production in grid-cell $j$ calculated based on the GLIMS livestock data (kg N/P km$^{-2}$ yr$^{-1}$
$^{1}$); $GPro_{x,c}$ is the GLIMS-based manure nutrient production at county $c$ where grid cell $j$ is
located (kg N/P yr$^{-1}$).
To generate grid-level manure production from 1860 to 1930, we obtained manure production
change rates (1860-1930) from the dataset developed by Holland et al. (2005) and applied them
to the grid-level manure nutrient production in 1930. Holland et al. (2005) provided global
annual manure N production data from 1860 to 1960. In order to combine this dataset with the
U.S. manure nutrient production data, we assumed manure production changes in the U.S. were
consistent with the global trend and the manure N:P ratio was constant during 1860-1930 (Zhang
et al., 2017).
**2.2 Manure nutrient application**
Manure nutrient application data were developed by allocating the county-level recoverable
manure nutrient according to the grid-level manure nutrient demand of crops. The recoverable
manure nutrient represents the proportion of manure nutrients that could reasonably be expected
to be collected from the confinement facility and later be applied to the land (Kellogg et al.,
2000). The recoverable manure nutrient was applied to cropland and pastureland according to
their demands. We calculated recoverable manure nutrient amounts by adjusting the county-level
manure production with recoverability factors provided by the Nutrient Use Geographic
Information System (NuGIS, http://nugis.ipni.net/). The nutrient demand was estimated
according to the assimilative capacity, the maximum amount of manure nutrient application
without building up nutrient level in the soil over time (Kellogg et al., 2000). We obtained the
proportion of recoverable manure nutrient that can be applied to cropland by combining the
assimilative capacities and areas of cropland and pastureland. The areas of cropland and
pastureland during 1860-2016 were derived from the HYDE 3.2 (Klein Goldewijk et al., 2017).
The above-mentioned processes are represented by the following equations:
$$APP_{x,c} = Pro_{x,c} \cdot Rf_{x,c} \cdot f_{x,crop,c} \tag{3}$$
$$f_{x,crop,c} = \frac{A_{crop,c} \cdot S_{x,crop}}{A_{crop,c} \cdot S_{x,crop} + A_{past,c} \cdot S_{x,past}} \tag{4}$$
where, $APP_{x,c}$ is the recoverable animal manure nutrient available for application on cropland in
county $c$ (kg N/P yr$^{-1}$); $Rf_{x,c}$ is the manure nutrient recoverability rate (the recoverability rates
are unitless and county-specific with average values 0.19 for N and 0.35 for P), and $f_{x,crop,c}$
refers to the fraction of manure nutrient that is available for cropland (unitless); $A_{crop,c}$ and
$A_{past,c}$ represent the annual area of cropland and pasture in each county (km$^2$), respectively,
while $S_{x,crop}$ and $S_{x,past}$ represent the average assimilative capacities of cropland and
pastureland (kg N/P km$^{-2}$), respectively (Table S2).
To spatialize county-level recoverable animal manure nutrient to gridded maps, we first
developed annual grid-level crop nutrient demands data as the base maps (Eq. 5). Nutrient
demands of crops were estimated by combining the assimilative capacities, harvested areas and
yields of 13 crops (maize, soybeans, sorghum, cotton, barley, wheat, oats, rye, rice, peanuts,
sugar beets, tobacco, and potatoes). The grid-level average assimilative capacity of cropland was
calculated based on crop-specific yield and harvested area maps in 2000 provided by Monfreda
et al. (2008). Next, this map of cropland assimilative capacity was integrated with dynamic
cropland fraction data (Yu and Lu, 2018) to obtain annual nutrient demand maps from 1860 to
2016. Original cropland fraction data was at a resolution of 1 km in the projected coordinate
system which approximates 30 arc-second resolution in the geographic coordinate. We resampled
the cropland fraction maps into the resolution of 30 arc-second to match the manure nutrient
production data.

$$Dem_{x,j} = \sum_{k=1}^{m} Y_{j,k} S_{x,k} \cdot Den_j \tag{5}$$

where, $Dem_{x,j}$ is the crop demand for manure nutrient in grid $j$ (kg N/P km$^{-2}$ yr$^{-1}$); $Y_{j,k}$ is the
yield of crop $k$ (ton per area of cropland) and $S_{x,k}$ is the manure nutrient assimilative of crop $k$
(kg N/P per ton product) (Table S3); $Den_j$ represents the cropland density in each grid (unitless).
The downscaling of county-level recoverable manure nutrient data into grid maps was similar to
the method used in developing manure nutrient production data (Eq. 2). The grid values on
manure nutrient demand maps were adjusted to match annual county-level recoverable manure
nutrient data (Eq. 6). The manure nutrient application data from 1860 to 2017 were developed
through the above-mentioned processes, however, several variables and parameters in these
processes were not available through the whole study period (e.g., manure recoverability rate,
crop yield). Therefore, we assumed these variables or parameters did not change before or after
the data-available period (Kellogg et al., 2000; Puckett et al., 1998).
$$APP_{x,j} = Dem_{x,j} \cdot \frac{APP_{x,c}}{Dem_{x,c}} \tag{6}$$

where, $APP_{x,j}$ is the manure nutrient application in grid $j$ (kg N/P km$^{-2}$ yr$^{-1}$) and $Dem_{x,c}$
refers to the demand for manure nutrients in county $c$ where grid $j$ is located (kg N/P yr$^{-1}$).

## 3 Results

### 3.1 Temporal and spatial patterns of manure nutrient production

We estimate that the total manure N and P production increased from 1.4 Tg N yr$^{-1}$ and 0.3 Tg P yr$^{-1}$ in 1860 to 7.4 Tg N yr$^{-1}$ and 2.3 Tg P yr$^{-1}$ in 2017, respectively (Fig 2). The manure N and P production reached the first peak in 1975 (6.1 Tg N yr$^{-1}$ and 1.8 Tg P yr$^{-1}$), and slightly declined thereafter, then regrew since 1987 with the second peaks occurring in 2007 (N) and 2017 (P), respectively. The slight decrease in manure nutrient production between 2008 and 2012 may be associated with the financial crisis and the low demand for livestock products. The total manure N and P production increased 5-fold and 7-fold during 1860-2017, with the increasing rates of 0.03 Tg N yr$^{-2}$ and 0.006 Tg P yr$^{-2}$ during 1860-1930, 0.05 Tg N yr$^{-2}$ and 0.02 Tg P yr$^{-2}$ during 1930-2017 (p<0.01), respectively. The N:P ratio in total manure production changed from 4.33 in 1930 to 3.25 in 2017. The decrease in the N:P ratio in total manure production was related to the change in the structure of animal population. For example, the proportion of beef cows and broilers (N:P ratio in excretion: 3.0-3.2) increased while that of milk cows and horses (N:P ratio in excretion: 5.5-6.7) decreased over the study period.

The spatial pattern of animal manure N and P production showed a similar change over the study period (Fig 3). The distribution maps showed that the Midwestern U.S. (e.g., Iowa, Missouri, and Illinois) was the core region (> 300 kg N km$^{-2}$ yr$^{-1}$ or 100 kg P km$^{-2}$ yr$^{-1}$) of manure N (P) production in 1860. From 1860 to 1930, the high manure nutrient production region (> 600 kg N km$^{-2}$ yr$^{-1}$ or 200 kg P km$^{-2}$ yr$^{-1}$) mainly enlarged outwards from the Midwest. Between 1930 and 1980, manure N (P) production not only intensified in the Midwest but also in the Southern U.S. (e.g., Texas, Georgia, and North Carolina). After 1980, manure N (P) production became more concentrated in many hot spots (> 6000 kg N km$^{-2}$ yr$^{-1}$ or 3000 kg P km$^{-2}$ yr$^{-1}$), especially in the

southeastern U.S. Meanwhile, part of regions around these hot pots experienced a decline in
manure production.
According to the change rates of manure nutrient production from 1860 through 2017 (Fig.4),
several growth poles (change rates > 20 kg N $km^{-2}$ $yr^{-2}$ or 5 kg P $km^{-2}$ $yr^{-2}$, $p<0.01$) located in
Iowa, Arkansas, California, Alabama, Pennsylvania were identified. The belt (change rates > 5
kg N $km^{-2}$ $yr^{-2}$ or 1 kg P $km^{-2}$ $yr^{-2}$, $p<0.01$) from Minnesota to Texas, as well as scattered areas
along the east and west coasts, were the primary contributors to the increase in manure N (P)
production. Aside from the huge increase in the Midwest and Southeast, decreasing trends were
exhibited in some regions, particularly the northeastern border of the U.S.
**3.2 Comparison of manure nutrient demand and production**
We assumed that the capacity of crops to assimilate nutrients was equal to manure nutrient
demand. From 1860 to 1930, the manure N (P) demand of cropland intensified and enlarged
inside the Corn Belt region (e.g. Iowa, Illinois, Minnesota, Nebraska, North Dakota, and South
Dakota), as well as the Southern U.S. (e.g., Texas, Oklahoma, Arkansas, Mississippi, Alabama,
Georgia, and Tennessee) (Fig 5). After 1930, change in the spatial pattern of manure nutrient
demand was dominated by the abandonment of cropland, and the magnitude of demand slightly
decreased, especially after 1980. Compared to the spatial patterns of manure production and
demand (Figs 3 and 5), it is worth noting that the high manure production and demand regions
overlapped in the Midwest and Southeastern U.S., but a large deficit (demand higher than
production) existed along the Lower Mississippi River Valley.
**3.3 Temporal and spatial patterns of manure nutrient application in cropland**
Animal manure N (P) application amount is primarily dominated by production and its spatial
pattern is impacted by demand. The overall manure application to production ratios were 0.15
and 0.23 for N and P, respectively. Driven by cropland expansion and enhanced manure
production, total manure N and P application in croplands increased 9-fold and 10-fold since
1860, reaching 1.3 Tg N yr$^{-1}$ and 0.6 Tg P yr$^{-1}$ in 2017 (Fig 6). The N:P ratio in manure
application decreased from 2.62 to 2.32 during 1930-2017. The substantial increase of manure N
and P application mainly happened in two periods: 1924-1970 (increase rates: 0.009 Tg N yr$^{-2}$
and 0.005 Tg P yr$^{-2}$, *p*<0.01) and 1987-2017 (increase rates: 0.01 Tg N yr$^{-2}$ and 0.005 Tg P yr$^{-2}$,
*p*<0.01). The variations of total application and production quantities didn't follow the same
trajectory. For example, from 1975 to 1987, when manure N production decreased, the total
manure application still remained stable. The application to production ratios reached the first
peak in 1891 (N: 0.14, P: 0.25) followed by a decrease until 1945 (N: 0.13, P: 0.20), and then
resumed the increasing trend through 2017 (N: 0.18, P: 0.25).
The spatial shift of manure application, similar to manure nutrient demand, gradually expanded
inside the Corn Belt and toward the Southern U.S. (Fig 7). The expansion of manure application
region primarily occurred during 1860-1930, induced by cropland expansion. After 1930, the
changed spatial patterns of manure N (P) application were characterized by intensified
application in the Midwest and multiple hot spots (> 2000 kg N km$^{-2}$ yr$^{-1}$ or 1000 kg P km$^{-2}$ yr$^{-1}$).
The spatial distribution of hot spots on application maps was similar to that on manure nutrient
production maps. In 2017, high manure nutrient application regions (> 500 kg N km$^{-2}$ yr$^{-1}$ or 200
kg P km$^{-2}$ yr$^{-1}$) mainly distributed in the Midwestern U.S., Southern U.S., Mid-Atlantic (e.g.,
Pennsylvania, Maryland, and Virginia), and California, where abundant recoverable manure
nutrients were applied in the local cropland to meet the high nutrient demand of crops. A quite
low manure nutrient application rate (< 100 kg N km$^{-2}$ yr$^{-1}$ or 50 kg P km$^{-2}$ yr$^{-1}$) was observed in
regions with less cropland demand (e.g., Southwestern U.S.) and low manure production (e.g.,
Lower Mississippi River Valley).
**3.4 Manure production and application across the major river basins**
From the 1860s to the 1970s, all basins exhibit increased manure nutrient production (Figs 8a
and 8b). However, from the 1970s to the 2010s, the manure N and P production decreased in the
New England and Missouri basins, while a dramatic increase was shown in the South Atlantic-
Gulf, Mid-Atlantic, and Arkansas-White-Red basins. Manure application demonstrated a similar
pattern across different basins (Figs 8c and 8d), but it increased from the 1970s to the 2010s in
most basins except the two basins (the New England and Souris-Red-Rainy) in the northern
regions. In the 1970s, the Missouri basin was the largest source contributing ~20% N (P) of the
total manure production, while the Upper Mississippi basin had the highest manure N (P)
application in cropland accounting for 19% N and 24% P of the total manure N (P) application.
During 2011-2017, however, the dominant regions of manure nutrient production and application
were shifted to the South Atlantic-Gulf basin which accounted for the largest single share (18%
N and 19% P of the total N (P) production, 24% N and 21% P of the total N (P) application). The
uneven distribution of manure application intensified during 1860-2017, demonstrated by the
standard deviation of manure N and P application across all basins consistently increasing from
0.013 Tg N yr$^{-1}$ and 0.005 Tg P yr$^{-1}$ in the 1860s to 0.081 Tg N yr$^{-1}$ and 0.038 Tg P yr$^{-1}$ during

311    2010-2017.

# 4 Discussion


**4.1 Comparison with previous investigations**
Within this study, we compared manure nutrients production data with other four datasets from
Food and Agriculture Organization Corporate Statistical Database (FAOSTAT, 2019), NuGIS,
Kellogg et al. (2000), and Yang et al. (2016). FAOSTAT provides total manure N production at
the national level from 1961 to 2017, while the other three datasets provide county-level manure
N and P production data. The estimated manure N (P) production from this study was lower than
the other two datasets (FAOSTAT and Yang et al.) before 1982 and started to become the highest
dataset after 2003 (Fig 9). During 1982-2007, the estimation from this study is very close to
other estimations developed at the county-level. The average total manure N (P) production over
1987-1997 was 6.02 Tg N yr$^{-1}$ (1.79 Tg P yr$^{-1}$), 6.75 Tg N yr$^{-1}$, 5.96 Tg N yr$^{-1}$ (1.75 Tg P yr$^{-1}$),
5.64 Tg N yr$^{-1}$ (1.67 Tg P yr$^{-1}$), and 6.01 Tg N yr$^{-1}$ (1.86 Tg P yr$^{-1}$), respectively, for this study,
FAOSTAT, NuGIS, Kellogg et al., and Yang et al. The differences among the datasets were
derived from calculation methods, chosen livestock types and numbers, as well as parameters,
such as animal-specific excreted manure nutrient rates and the number of days in the life cycle.
In terms of changing trends, manure N and P production were relatively stable after the 1960s in
FAOSTAT, Kellogg et al., and Yang et al., while the NuGIS data increased slightly between 1987
and 2007 and then decreased sharply after 2010. In contrast, our results showed an increasing
trend after the 1980s due to the consideration of the increased animal body sizes.
In the previous four datasets, temporal changes in manure N (P) production are driven by animal
numbers. It is worth noting that manure N (P) production can still increase despite the
stabilization of livestock numbers in recent years. Driven by the advanced technology, livestock
live weight and size consistently increased, which may enhance the manure nutrient excretion
rate of each animal (Lassaletta et al., 2014; Sheldrick et al., 2003; Thornton, 2010). We
compared manure nutrient production calculated with constant average weights and with
dynamic weights of livestock. The results showed that manure production with dynamic weights
increased dramatically after the 1990s (Fig.10). Enhanced livestock weights contributed 59% and
54% of the increase in manure N and P production, respectively, from 1987 to 2017 when the
differences between the two total production data reached 0.98 Tg N yr$^{-1}$ and 0.31 Tg P yr$^{-1}$.
It is difficult to compare our dataset of manure N (P) application in soils with previous studies
since these datasets provided reference values with various definitions and were generated based
on different statistical methods. For example, FAOSTAT provided annual data of "Manure
applied to soils" in the U.S., whereas this dataset was developed based on the assumption that all
treated manure, net of losses (e.g., $NH_3$ volatilization, N leaching, and runoff), is applied to soils
following the method in the 2006 IPCC guidelines (Eggleston et al., 2006). Kellogg et al. (2000)
and NuGIS both estimated recoverable manure nutrients by multiplying confined livestock units,
recoverability factors, and nutrients per ton of manure after losses. All three datasets do not
separate manure application to cropland and pastureland. This study developed manure nutrient
application data in cropland by applying the method of recoverability factor in combination with
the cropland nutrient assimilative capacity. Compared to the other three datasets, our data
subtracted the proportion of manure application on pastureland and considered the impact of the
change in cropland area, which can lead to relatively low data values.
**4.2 The impact of manure nutrient enrichment on coastal oceans**
Animal manure N (P) that is lost through surface runoff or leaching exacerbated eutrophication
and hypoxia in the aquatic system in the U.S. (Feyereisen et al., 2010; Williams et al., 2011).
During the expansion of manure production from the Midwest to the Southeastern coastline,
massive amounts of nutrients get more of a chance to be transported to the estuary. When rivers
transport nutrients from land to coastal oceans, nutrients could be removed or retained through
denitrification, plant and microbial uptake, organic matter burial in sediment, and sediment
sorption (Billen et al., 1991; Seitzinger et al., 2002). As the accumulated manure gets closer to
the coastline, manure nutrients that enter into rivers may be less likely to decrease during
transportation due to the short distance. Additionally, the risk of massive manure loss in
hurricane events increases under the background of enhanced Atlantic hurricane activities since
1995 (Saunders and Lea, 2008; Trenberth, 2005). Flooding rains and high winds may destroy
manure storage structures (e.g., pad, pond, lagoon, tank, and building), resulting in the direct
release of untreated manure into rivers (Tabachow et al., 2001).
The South Atlantic-Gulf and Mid-Atlantic basins are two critical coastal regions with the
enrichment of manure nutrient production and application from the 1970s to the 2010s due to
intensive livestock farming. The low recovery and reuse rate of animal manure N (P) can
potentially cause a significant amount of manure N and P exports from the basins into the Gulf of
Mexico and the Atlantic Ocean (Sheldrick et al., 2003). The Upper Mississippi, Missouri, and
Arkansas-White-Red sub-basins within the Mississippi River basin were the three largest sources
of manure production and were the dominant contributors to N and P loads into the Gulf of
Mexico (David et al., 2010; Jones et al., 2019). The Upper Mississippi and Missouri basins had
the highest manure nutrient production and application in the 1970s and maintained the high
quantities until 2010, while manure N (P) production and application largely increased in the
Arkansas-White-Red basin during 2011-2017. The enhanced total manure production may
continually be responsible for the enriched loads of N and P that can lead to coastal water
pollution (Rabalais and Turner, 2019).
**4.3 Implication for manure nutrient management**
The structure of animal agriculture has shifted toward the concentrated animal feeding operations
(CAFOs), which led to the increased numbers of animals in confinements (Kellogg et al., 2000).
Thus, manure production became increasingly concentrated in several regions with large
operations. Meanwhile, the decreased manure production in partial areas of the Midwestern and
Southern U.S was due to the disappearance of small family farms. On the other hand, the
enhanced animal weight caused an additional increase in manure production in operations with
plenty of confined animals. The unevenly intensified distribution of manure production may have
further exacerbated the imbalance of regional nutrient allocation. Currently, opportunities for
widespread manure application are limited because the transport of manure can be costly.
Furthermore, the long distance between livestock farms and cropland can bring difficulties to
practical operations (MacDonald, 2009). There remain gaps between manure production and
demand in some regions of the U.S. (e.g., the Lower Mississippi River Valley). In contrast,
manure collected from many farms cannot be properly used to fertilize crops. The unusable
manure is not only a waste of manure resources, but may also cause serious environmental
problems through nutrient losses into the atmosphere and aquatic systems.
The efficient recovery and processing of manure nutrients, the transportation of manure from the
CAFOs to the specific crop area, and the utilization of manure as bioenergy can be important
pathways to control pollution caused by the uneven distribution of manure production (He et al.,
2016). The CAFOs facilitate the recovery of animal manure, which has created conditions for
large-scale utilization and management of manure. Because of the economies of scale, the cost of
transportation and management for per unit animal manure can be reduced, making the
utilization of manure more feasible. Establishing a direct link between CAFOs and specific crop
area ensures that animal manure production can be consumed in large quantities and thereby
improving economic efficiency. For the centralized management of animal manure, nutrient
losses during collection, storage, and application should be constrained or avoided, because a
small proportion of nutrient losses can even contaminate regional environment if manure nutrient
amounts are huge. Manure management systems with the integrated package of measures are
necessary for controlling nutrient losses from the feed–animal–manure–crop chain (Oenema et
al., 2007).
**4.4 Assumptions and Uncertainties**
Uncertainties in this study are primarily associated with data sources and methods that were
used. First, multiple data sources were used to develop the datasets of manure production and
application data; however, biases exist in these source data. For instance, the non-disclosure of
the livestock data in the USGS Census of Agriculture can cause an underestimate of manure
production in numerous counties (Yang et al., 2016). Second, the parameters in the calculation
model, e.g., excreted manure nutrient rates, could bring uncertainties in the estimation of animal
manure nutrient production and application. Third, various assumptions were made in this study
to extend the time series of data and spatialize data from the county-level to the grid-level. These
assumptions were established based on available data and experience, but uncertainties still
existed and influenced the accuracy of the dataset. The limitations and uncertainties of these
assumptions were further discussed and explained in the following part.
The livestock distribution maps from the GLIMS dataset were the reference of the spatial pattern
of manure nutrient production data within each county. The GLIMS data were developed by
establishing statistical relationships between livestock inventory data and multiple environmental
variables (e.g., climate, land cover, human activities), and using these relationships to predict
livestock distributions across the globe. We assumed livestock distribution within each county
was stable over the study period because the dynamic livestock maps were unavailable.
However, the environmental variables can change and induce the variation in livestock

430 distribution inside each county. The accuracy of this manure nutrient production dataset can be

431 improved once dynamic livestock maps are developed in the future.

432 The manure nutrient production before 1930 was generated based on change rates in global

433 manure N datasets from Holland et al. (2005). There is a period of overlap (1930-2004) between

434 this global dataset and the USDA census data. During 1930-2004, the average annual change

435 rates of manure N production were 1.08% in the global dataset and 1.01% in this study.

436 Therefore, the changes in estimated manure N production in the U.S. before 1930 might be

437 reasonable at a long-time scale. The ratio of N to P in animal manure varies among different

438 animal species and changes along with proportions of different animal populations over time.

439 From 1930 to 2017, the N:P ratio in the total manure production slightly decreased from 4.33 to

440 3.25. Due to the lack of manure P production data before 1930, we calculated manure P

441 production in this period according to manure N production and the constant N:P ratio in 1930. If

442 the N:P ratio kept decreasing before 1930, the total manure P production may be overestimated

443 during 1860-1929.

444 Changes in recoverability factors and crop yields over the study period were ignored due to lack

445 of data support and that may cause a bias in quantifying manure nutrient application. With the

446 development of livestock confinement facilities, the confinement and recoverability factors of

447 animal manure may increase in recent decades (Kellogg et al., 2000). Hence, manure application

448 can be overestimated before the 1980s and underestimated after the 2000s. The yields of

449 different crops may change at different speeds over the study period, and that can affect the

450 spatial patterns of manure nutrient demand of cropland as well as manure nutrient application.

451 In addition, the development of manure application data was based on two assumptions: (1) The

452 allocation of manure nutrient application within the county was proportional to crop nutrient

demands; (2) Manure is assumed to be applied in the county where it was produced. Manure
application is controlled by distance, cost, and operating practice of humans. Currently, the
specific locations of animal farms across the country are not available, thus it is difficult to
evaluate the influence of distances between farms and croplands. Due to the practical limits of
manure transportation (Buckwell and Nadeu, 2016; MacDonald, 2009), it is reasonable to
assume manure production and application happen within the same county on a large scale.
However, ignoring the impact of multiple factors on manure application within the county can
still result in biases in the spatial distribution of manure application.

## 5 Data availability

The gridded datasets of manure N and P production and application in the contiguous U.S. are
available at https://doi.org/10.1594/PANGAEA.919937 (Bian et al., 2020). A supplement is
added to provide information about manure demand and all parameters used to develop the
datasets.

## 6 Conclusion

Manure nutrient production and application in the livestock-crop system substantially altered the
regional and global N and P cycle. In this study, we developed geographically explicit datasets of
animal manure N and P production and their application in cropland across the contiguous U.S.
from 1860 to 2017. The dataset indicated that both manure N and P production and application
significantly increased over the study period. Although livestock numbers became stable in
recent decades, manure nutrient production still increased due to the enhanced livestock body
weight after the 1980s. Enhanced livestock weights contributed 59% and 54% of the increase in
manure N and P production, respectively, from 1987 to 2017. Meanwhile, manure nutrient
production intensified and enlarged inside the Midwest and toward the Southern U.S. from 1980
to 2017, and became more concentrated in numerous hot spots. As manure nutrient application
also expanded toward the Southeastern coastline, massive amounts of nutrients get more of a
chance to be transported to the estuary. The enrichment of manure nutrients in the South
Atlantic-Gulf, Mid-Atlantic, and Mississippi River basins increased the risk of excessive nutrient
loading into the Gulf of Mexico and the Atlantic Ocean under extreme weather conditions.
Therefore, it is of great importance to effectively store, utilize, and transport animal manure in
order to reduce nutrient pollution and restore the environment.

**Author contributions**
HT designed and led this work. ZB is responsible for developing the datasets. QY provided the
county-level livestock dataset. RX proposed the methods in the study. SP and BZ analyzed the
results. All authors contributed to the writing of the manuscript.
**Competing interests**
The authors declare that they have no conflict of interest.
**Acknowledgments**
This study has been supported in part by National Science Foundation grant (1903722); National
Oceanic and Atmospheric Administration grants (NA16NOS4780204, NA16NOS4780207); the
National Aeronautics and Space Administration grants (NNX12AP84G, NNX14AO73G,
NNX10AU06G, NNX14AF93G), and OUC-AU Joint Center Program.

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

Table 1. Summary of data sources

| Data variables | Time period | Resolution | Reference/source |
|---|---|---|---|
| Livestock numbers | 1930-2017 | County-level | USDA National Agricultural Statistics Service https://www.nass.usda.gov/index.php |
| Livestock weights | 1921-2017 | Country-level | USDA Economic Research Service database http://www.ers.usda.gov/ |
| Livestock distribution | 2007 | 30 arc-second | Global Livestock Impact Mapping System (GLIMS) (Robinson et al., 2014) |
| Manure recoverability rates | 1987-2014 | County-level | Nutrient Use Geographic Information System (NuGIS) http://nugis.ipni.net/ |
| Crop harvested area and yield | 2000 | 5 arc-min | (Monfreda et al., 2008) |
| Crop and pasture distributions | 1860-2016 | 5 arc-min | History Database of the Global Environment (HYDE 3.2) (Klein Goldewijk et al., 2017) |
| Crop density | 1850-2016 | 1×1 km | (Yu and Lu, 2018) |









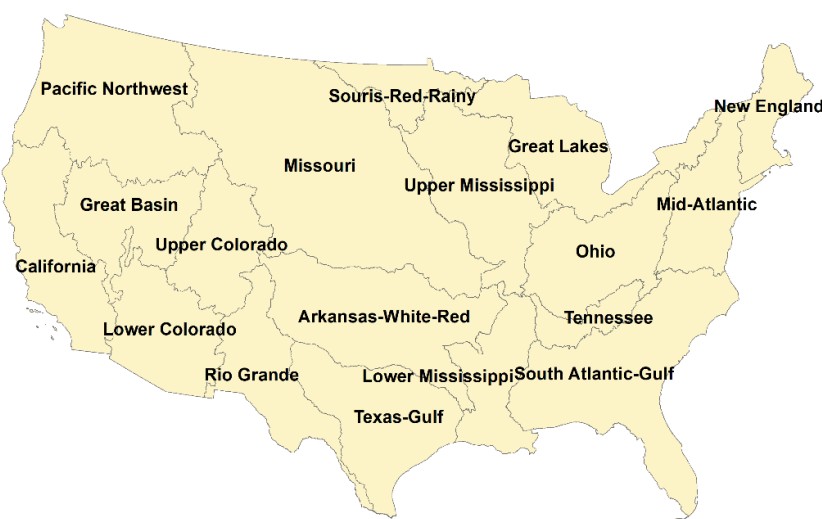


Figure 1. Eighteen Hydrologic Units in the contiguous U.S. (Recreated from the U.S. hydrologic unit
map: https://water.usgs.gov/GIS/regions.html)












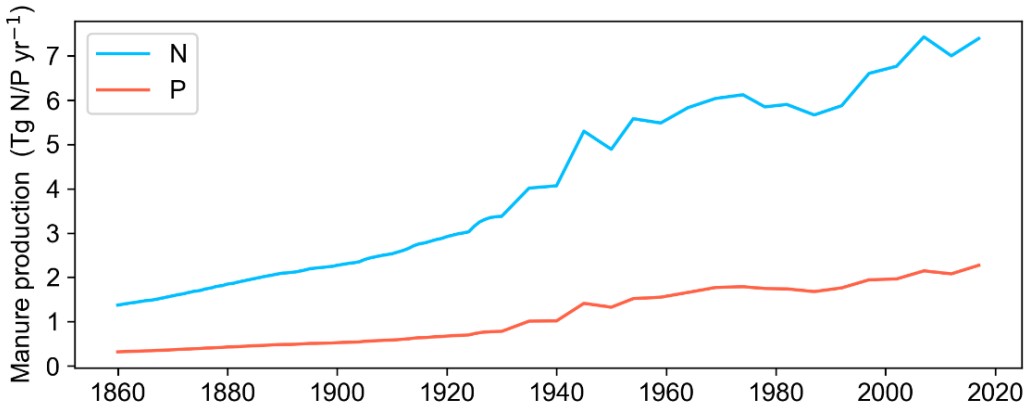


Figure 2. Trend and variation of total manure N and P production in the contiguous U.S from 1860 to
696 2017













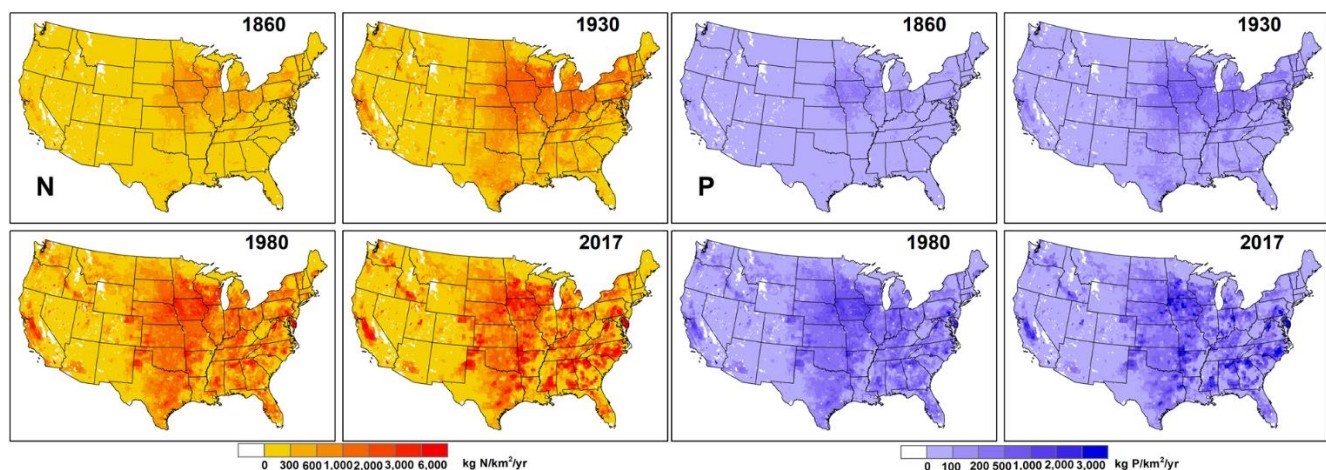

Figure 3. Spatial distribution of manure N and P production across the contiguous U.S. in 1860, 1930, 1980, and 2017. (Note: 1930 and 2017 were the earliest and latest years of available USDA census data, respectively, and 1980 was chosen as the year at the middle of these two years)

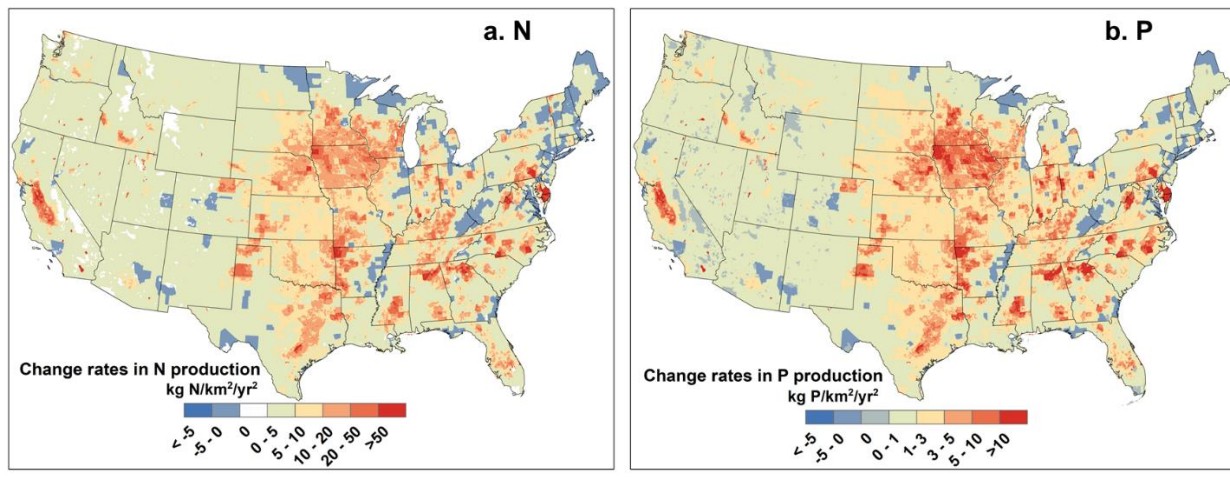


Figure 4. Change rates of manure (a) N and (b) P production across the contiguous U.S. during 1860-
726             2017. (Note: the increasing rates were calculated based on the Mann-Kendall Test)



















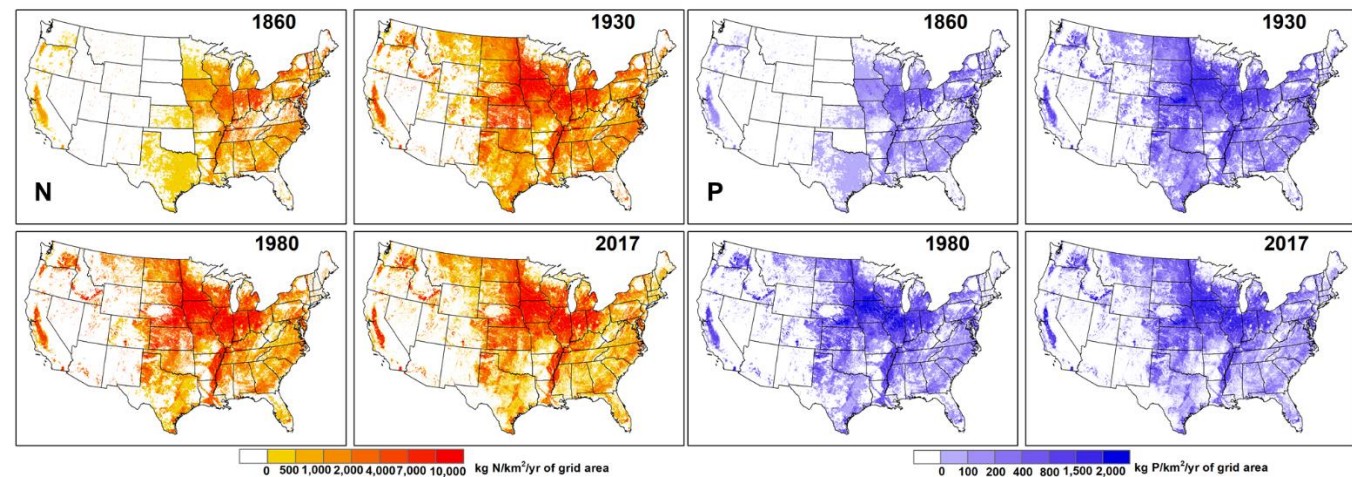

Figure 5. Spatial distribution of N and P demand of crops in 1860, 1930, 1980, and 2017.

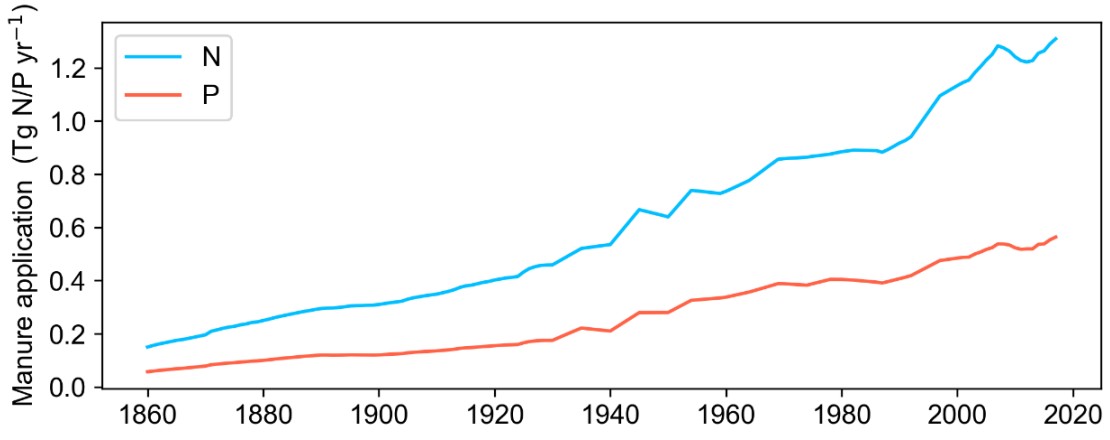


Figure 6. Trend and variations of total manure N and P application in the contiguous U.S. from 1860 to
760                                                                2017













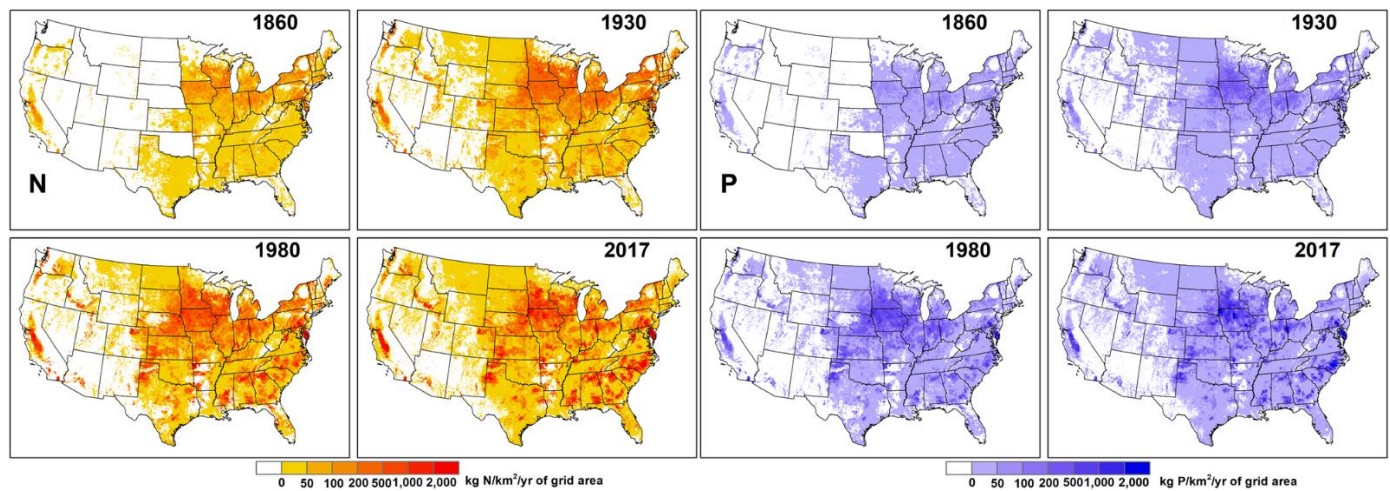



Figure 7. Spatial distributions of manure N and P application in the U.S. cropland in 1860, 1930, 1980,
and 2017.








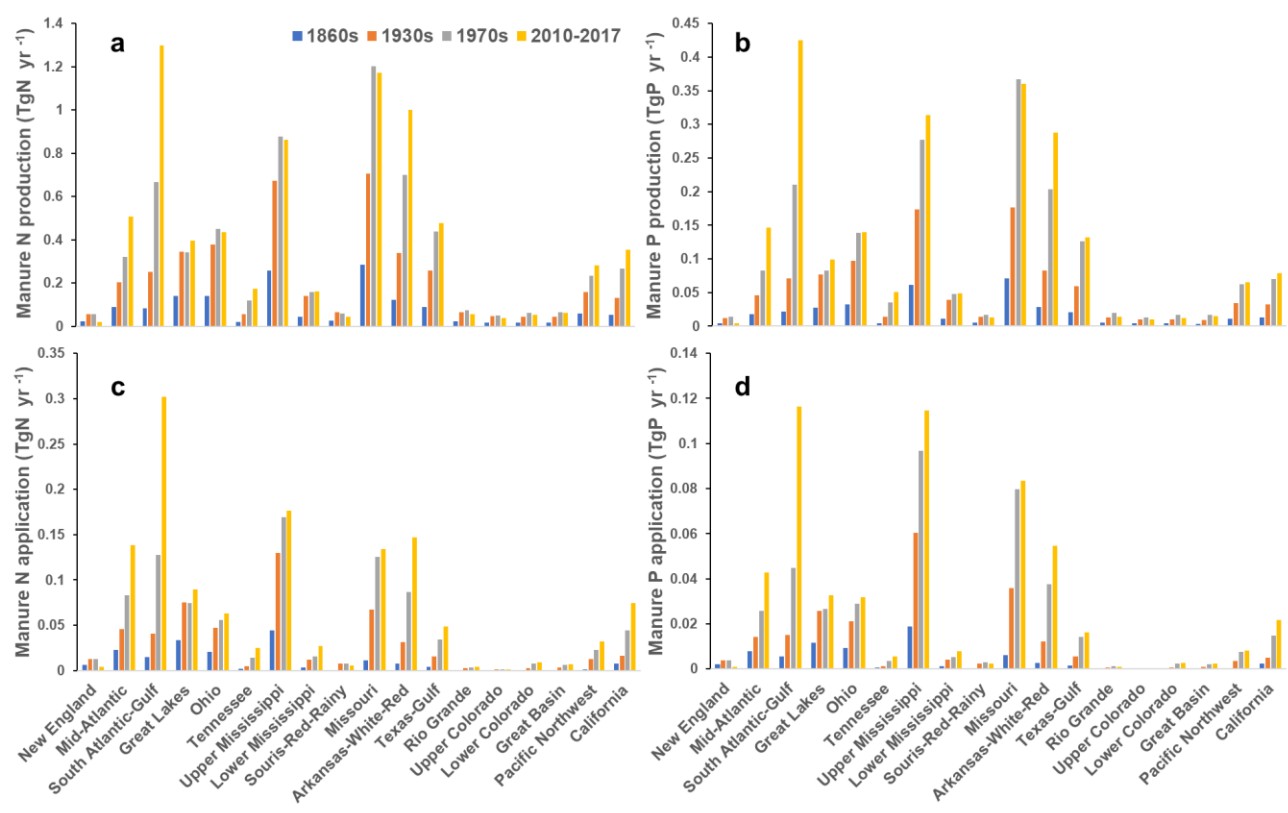

Figure 8. Average annual manure production (a. N, b. P) and application (c. N, d. P) across 18 major

basins in the 1860s, 1930s, 1970s, and 2010-2017.















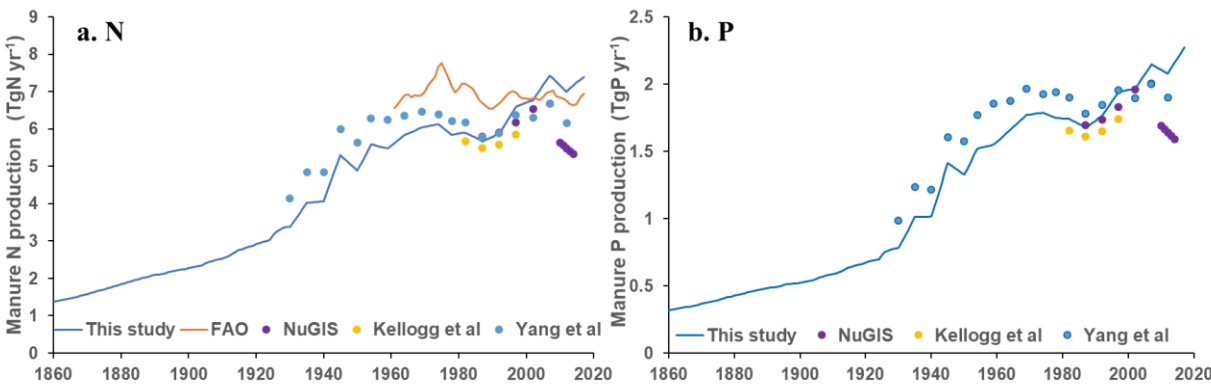


Figure 9. Comparison of manure nutrients production in this study with the four previous datasets.
















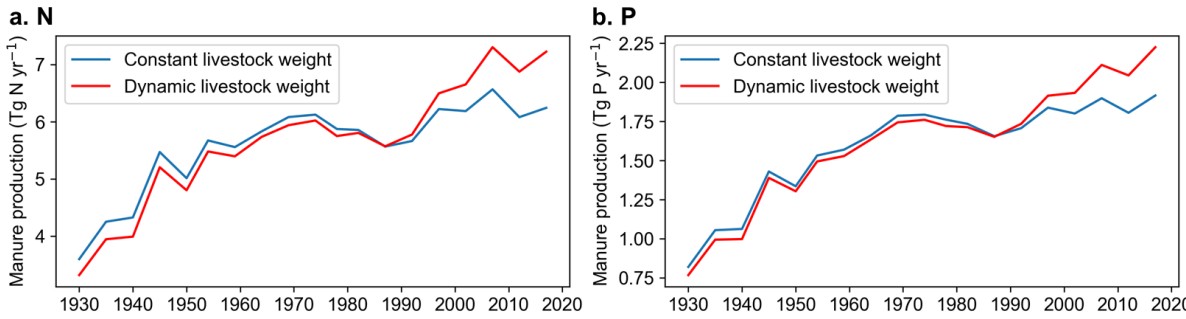


Figure 10. Comparison of manure N (P) production calculated based on dynamic weight of livestock and
819                                              constant weight.
