# Peer review of "Production and application of manure nitrogen and phosphorus in the United States since 1860"

_Earth System Science Data, 2020_

## Referee Comment (RC1) · Anonymous Referee #1 · 10 Nov 2020

The manuscript "Production and application of manure nitrogen and phosphorus in the United States since 1860" developed datasets of annual animal manure N and P production and application in the U.S. continuously at a 30 arc-second resolution over the period of 1860-2017. In general, the manuscript is well written. However, this study should pay more attention to quantitatively describing regional livestock manure nutrients status (surplus or deficit ) on the basis of manure production and application, which is of vital importance for regional nutrient management of U.S. I cannot recommend for publication in this prestigous journal because the innovation of this manuscript is not as strong. The following issues should be properly addressed. Abstract: The first two sentences are too long. It needs to be brief and clear enough to support the objective of the study. It should be rewritten concisely to define the aim of the study.

[Figure]

Introduction: The author should better indicate the scientific hypothesis and specific objectives of this study one by one in the last paragraph of Introduction. Methods: "We assumed that spatial pattern of livestock distribution inside each county were static over the study period" in row 126, "we assumed manure production changes in the U.S. were consistent with the global trend and manure N:P ratio was constant during 1860-1930" in row 146, "we assumed these variables or parameters did not change before or after the data-available period." in row 195. Related references should be provided. Results: Statistical analysis should be described in more details in this section. Any detailed regional data to make comparision possible? The first sentence of "3.1 temporal and spatial patterns of manure nutrient production"( in row 202), the author calculated the average total manure N and P production during 1860-2017. What's the meaning of this data in such a large time scale. Similar to the row 234 (the average annual manure N and P application in the U.S.). In row 204, please rewrite the sentence "Compared with production rates in 1860 (1.4 Tg N yr-1 and 0.3 Tg P yr-1 204 ), manure N and P production in 2017 increased 5-fold and 7-fold at a rate of 0.04 Tg N yr-1 and 0.01 Tg P yr-1 205 , respectively". It's better to note that 1.4 Tg N yr-1 and 0.3 Tg P yr-1 is manure N and P production. The reason why the author chose the year of 1860, 1930, 1980, and 2017 for analysis should be specified in the manuscript. In row 238, "total manure P application reached 2.3 Tg P yr-1 in 2017", please recheck the value of 2.3 Tg P yr-1 which does not match Fig.5. Discussion: In "4.1 Comparison with previous investigations", the authors have compared the results with four previous studies. So how to quantitatively describe the discrepancies caused by different coefficients or statistical methods? Data availability: Since the author have cited the data resource in Methods, this part can be moved to Methods. Using words/phrases: The language of this manuscript should be re-polished.

---

## Referee Comment (RC2) · Anonymous Referee #2 · 17 Nov 2020

This article presents a dataset about manure N and P production and use in the United States of America and its evolution in time. It is globally well written. However, my opinion is that it lacks of precision, discussion about main hypothesis and clarification of the main findings.

More particularly, in the introduction, nutrients in manure are mainly presented without considering nutrients in mineral fertilization while they are related with them by the bio-physical N and P cycles including the environmental consequences of losses of some of N and P components. In the introduction more precision is required, for example in definition and characterization of manure. (line 70 and followings). For example urea is not always the main N component and is not always readily usable by plant (toxicity for young plants). N composition of manure depend of many factors (excretion, . . .

[Figure]

fermentation dynamic and thus evolution during manure storage). There is confusion between excretion and manure after storage. So it is not always clear what is included within the term manure or N produced (N straw included?, only excreted? Before or after volatilisation?). I would also emphasize, in a circular approach, the notion of N and P source and sink at local/global level. There is a lack of clear identification of the mean objectives of the paper and the underlying hypotheses.

In the method section I would add a summary explaining the main steps as introduction of the method section. When information from other works or methods are used, like at line 120, i recommend to explain them briefly and add some summarizing values ( as for Line 166 manure nutrient recoverability rate ) Some hypothesis are done and should be better explained eg. Line 126 (spatial distribution of livestock). I recommend to use only SI metrics. Regarding the statistics presented I wonder if it would not be more appropriate to present them as median and percentiles. In the result and the discussion sections there is a need for better explanation of the results observed at local scale and their consequences. I would also expect discussion about the relation between P and N evolution if possible including mineral fertilization or biological fixation of nitrogen. I was also surprised by the difference for P between production and application. The fate of P and the losses have to be explained and validated. In the discussion section the main hypotheses like those of line 126 cattle distribution ot line 176 regarding evolution of crop yield over time have to be discussed. Can the peak and decrease in N production around 2008 be explained? I would add data in the section lines 280-285 and expect explanation about the observed differences. To me the conclusion is more a summary than a conclusion. It can also be improved.

References have to be added for all the statements like at line 53, 69 I would reformulate lines 54 to 60, 66 to 69, 211.

Line 174 maize instead of Maize Line 179 - =1? Line 203 (mean $\pm$ standard deviation) in the material and method section Lines 254 -256 in the material and method section Line 274 remove the "our" Line 282 Kellogg et al. (2000), and Yang et al. (2016) within

brackets

---

## Author Comment (AC1) · 23 Dec 2020

ESSD, December 23, 2020

Dear Editor:

Thank you for your letter and for the reviewers' comments concerning our manuscript entitled "Production and application of manure nitrogen and phosphorus in the United States since 1860" (ID: essd-2020-185). These constructive comments are valuable and helpful for improving our manuscript, as well as providing important guidance to our future research. We have addressed all comments raised by reviewers and provided point-to-point responses outlined in the Response to Reviewers enclosed. We have made the following major changes in the revised manuscript: 1. The Abstract

has been revised based on the reviewers' comments. 2. The Introduction has been rewritten to make our innovation and objectives clearer. 3. A summary has been added in the Method section. 4. More detailed information and analysis have been added in the Results section. 5. A discussion regarding the hypothesis has been added. 6. The Conclusion has been revised to clarify our key findings. We agree with the 1st Reviewer's comment on the importance of regional livestock manure nutrients status (surplus or deficit) for nutrient management in the U.S., but such a topic that requires extensive new effort is beyond the scope of the current manuscript. As we have described in the manuscript, the major objective of this study is to develop long-term, spatially-explicit manure nutrient data for the scientific communities for better investigating the nutrient budget and biogeochemical cycles at multi-scales across the U.S. Unfortunately such dataset is not available yet, which has limited our understanding and prediction of how manure nutrient affects crop production, greenhouse gas emission, and water pollution, etc. The gridded manure nutrient data developed in this study is the first manure nutrient dataset with a high spatial resolution (30 arc-second) covering one and a half century (158 years) across the conterminous U.S. We are confident that our datasets fill a critical data gap but urgently needed for a wide range of users to better understand and assess environmental challenges associated with manure nutrient enrichment. Hope the corrections and explanations we made can meet with your approval.

Thanks in advance for your consideration.

Sincerely,

Hanqin Tian Solon & Martha Dixon Professor Director of International Center for Climate and Global Change Research, Auburn University

Point-to-point responses to comments from reviewers

Reviewer 1 Comment 1: The manuscript "Production and application of manure nitrogen and phosphorus in the United States since 1860" developed datasets of annual

animal manure N and P production and application in the U.S. continuously at a 30 arc-second resolution over the period of 1860-2017. In general, the manuscript is well written. However, this study should pay more attention to quantitatively describing regional livestock manure nutrients status (surplus or deficit) on the basis of manure production and application, which is of vital importance for regional nutrient management of the U.S. I cannot recommend for publication in this prestigous journal because the innovation of this manuscript is not as strong. Response: We very much appreciate the issues and comments raised by the reviewer. The major objective of this study is to develop spatially-explicit manure nutrient data for the scientific community to better investigate the nutrient budget and biogeochemical cycles at multi-scales. Manure nutrient is one of the critical components in N and P cycles, and its production and application are dominated by human activities. Considering the importance of manure nutrient on crop production, greenhouse gas emission, and water pollution, it is a vital need to develop spatially-explicit datasets of livestock manure nutrient production and application. Doing so we can visualize hotspots of nutrient enrichment and the spatial shifts of environmental problems over time. Moreover, these datasets can serve as driving forcings for process-based ecosystem models, hydrological models, and machine learning models to investigate the dynamics of nutrient cycles under environmental changes. To the best of our knowledge, there have not been any previously developed manure production and application datasets at a spatial resolution of 30 arc-second covering more than 150 years from 1860 to 2017. We believe that our datasets have filled this critical data gap that has limited the predictive understanding of how increased manure production and application have affected biogeochemical cycles and water quality in the United States, which clearly falls into the scope of the journal Earth System Science Data (ESSD). More specific details of innovative work in this manuscript have been clarified as follows: 1. This study provides spatially explicit manure nutrient datasets over a time period of more than 150 years (1860-2017). Currently, most studies only provided county-level manure nutrient production data in the U.S., and these publicly available manure nutrient data cannot cover the period before the 1980s. Nevertheless, terrestrial biosphere models require spatially explicit manure nutrient input data to simulate the anthropogenic effect on biogeochemical cycles since the pre-industrial period. Studies focusing on soil nutrient storage change and legacy soil nutrient also need long-time manure nutrient data. Our dataset is not only the first manure nutrient data that updates to 2017 based on the latest USDA census report, but also can provide manure nutrient information back to 1860. It was difficult to collect county-level livestock information from the earlier version of the USDA census report, especially the first four reports (1930–1945). We manually digitalized these data (livestock information in more than 3000 counties) from the printed USDA reports, which involved a large amount of work. Therefore, this century-long scale manure nutrient dataset is very important and necessary for the communities. Actually, we have received many requests from a wide range of users who would like to use our datasets for different purposes. 2. The impact of dynamic livestock weight was considered when calculating manure nutrient production. Previous studies usually assumed that livestock weight was constant over time when quantifying nutrient production based on livestock number, and the impact of livestock weight on manure production was not considered. In this study, we found that although livestock numbers became stable in recent decades, manure nutrient production still increased due to the enhanced livestock body weight after the 1980s, which was not caught by other studies. Enhanced livestock weights contributed 59% and 54% of the increase in manure N and P production, respectively, from 1987 to 2017. 3. We developed manure N and P application in cropland datasets by integrating manure nutrient production and nutrient demand of crops. Manure nutrient application, as the direct nutrient input for the soil-crop system, maybe more important in nutrient budget compared with production. However, we noticed that many studies only use manure production data as input to estimate the contribution of manure nutrient on water quality, and one of the reasons is that spatially-explicit manure nutrient application dataset is still scarce. Geographically explicit manure nutrient application in cropland (excluding pasture) has rarely been estimated across the U.S. Manure nutrient can be applied to both cropland and pastureland, and other studies didn't separate them. This

study developed manure application data on cropland and considered the change in cropland over the study period. In order to develop this manure nutrient application data, we comprehensively considered the balance between manure production and demand. And this method was first proposed by us. Our manure nutrient application data can help other studies more accurately estimate the manure nutrient input in cropland. 4. We found that the South Atlantic-Gulf and Mid-Atlantic basins were exposed to high environmental risks due to the enrichment of manure nutrient production and application from the 1970s to 2010s. The distribution of manure nutrient production became increasingly concentrated in several regions, including the southeastern U.S. During the expansion of manure production and application from the Midwest to the Southeastern coastline, massive amounts of nutrients get more of a chance to be transported to the estuary. The enrichment of manure nutrient in the South Atlantic-Gulf, Mid-Atlantic, and Mississippi River basins increased the risk of excessive nutrient loading into the Gulf of Mexico and the Atlantic Ocean.

We agree with your suggestion that it is interesting to quantitatively describe manure nutrient status on the basis of manure production and application. The manure nutrient application in this study was estimated based on manure nutrient production and manure nutrient demand of cropland. Specifically, the amount of manure nutrient application within each county was mainly controlled by manure production and recoverability factors, while the spatial distribution of manure nutrient application within each county was consistent with the spatial pattern of manure nutrient demand of cropland. The recoverability rates are county-specific with average values 0.19 for N and 0.35 for P, therefore, the manure nutrient application was largely lower than manure nutrient production (Fig.1). Meanwhile, we developed manure nutrient demand dataset based on crop nutrient assimilative capacities. For the contiguous U.S., the total demand for manure N was higher than the production over the study period while the total production of manure P started to exceed demand after the 1980s (Fig.1). The gap between production and demand has considerably narrowed since the 1920s due to the cease of the increase in cropland area. More details can be found based on the distribution

maps of manure nutrient production, demand, and application in the manuscript and supplementary. Since the estimation of manure nutrient application relied on the manure nutrient production, the difference between these two variables may mainly be derived from the parameters. Furthermore, the nutrient demand of cropland was not only satisfied by manure nutrient, but also by fertilizer and biological fixation. To investigate the regional nutrient surplus or deficit in agricultural systems, it is better to include mineral fertilization, biological fixation of N as well as removed nutrient by harvested crop.

We did search literature and collect datasets of fertilizer and N fixation during the revision stage. Currently, most studies only provided county-level nutrient dataset after the 1980s. We have compared our manure data with the dataset from Nutrient Use Geographic Information System (NuGIS) (Tables 1 and 2). It showed that manure N and P application may account for a relatively small share compared with synthetic fertilizer. N and P balances demonstrated that N kept a surplus state and P fluctuated between surplus and deficit during 1987-2014 in the U.S. If we want to conduct this analysis at a long-time scale, we need to collect or develop fertilizer, N fixation, and nutrient in harvested crop datasets with the same spatial resolution and long-time series. Therefore, estimating nutrient status may involve huge amounts of work and even need efforts from other groups, which is certainly beyond the scope of this study. For the current study, we may focus on the manure nutrient, and we will definitely conduct the study of N and P budget across the U.S. in our future work. Meanwhile, the manure data developed in this study provide opportunities for all researchers to estimate regional nutrient status at a long-time scale. We believe more meaningful conclusions can be found based on a more robust independent study.

Comment 2: Abstract: The first two sentences are too long. It needs to be brief and clear enough to support the objective of the study. It should be rewritten concisely to define the aim of the study. Response: Thank you for pointing out these issues in the Abstract. We have rephrased it to make it more concise and clearer. The first two sentences have been changed to "Livestock manure nitrogen (N) and phosphorus (P) play an important role in biogeochemical cycling. Accurate estimation of manure nutrient is important for assessing the regional nutrient balance, greenhouse gas emissions, and water environmental risk. Currently, spatially-explicit manure nutrient datasets over century-long period are scarce in the United States (U.S.)." (L26-30)

Comment 3: Introduction: The author should better indicate the scientific hypothesis and specific objectives of this study one by one in the last paragraph of Introduction. Response: We have rewritten the last paragraph of the Introduction section according to your suggestion. "Considering the importance of manure nutrient on crop production, greenhouse gas emission, and water pollution, it is vital to have a better understanding of livestock manure production and application at national or even global scales (Potter et al., 2010; Sheldrick et al., 2002; Tian et al., 2016). Precise and spatialized manure nutrient data can benefit stakeholders from finding available local recyclable nutrient sources or making strategies to minimize N and P losses. Currently, most studies only provided county-level manure nutrient production data in the U.S., with relatively short periods (Kellogg et al., 2000; Ruddy et al., 2006). Nevertheless, terrestrial biosphere models usually require spatially explicit manure nutrient input data to simulate the anthropogenic effect on biogeochemical cycles since the preindustrial period (Tian et al., 2019). Studies focusing on soil nutrient storage change and legacy soil nutrient also need long-time series manure nutrient data (MacDonald et al., 2012; Rowe et al., 2016). Moreover, previous studies usually assumed that nutrient excretion per animal is constant over time when quantifying nutrient production based on livestock number, which may lead to uncertainties (Zhang et al., 2020). Geographically explicit manure nutrient application in cropland (excluding pasture), as the direct nutrient input for the soil-crop system, hasn't been specifically estimated across the U.S. In this study, our objectives are to (1) develop grid-level manure N and P production datasets in the U.S. based on county-level livestock populations, dynamic livestock weight over time, and high-resolution livestock distribution maps; (2) develop grid-level manure N and P application in cropland datasets by integrating manure nutrient production and nutrient

demand of crops; (3) investigate the spatiotemporal patterns of manure nutrient production and application based on these datasets, and (4) further identify regions with a high risk of excessive nutrient loading. The four datasets display the masses of manure N and P per area in each 30×30 arc-second grid-cell during 1860-2017. The datasets can be used to drive ecosystem, land surface, and hydrological models to simulate manure-induced greenhouse gas emissions and nutrient loadings.". (L89-112)

Comment 4: Methods: "We assumed that spatial pattern of livestock distribution inside each county were static over the study period" in row 126, "we assumed manure production changes in the U.S. were consistent with the global trend and manure N:P ratio was constant during 1860-1930" in row 146, "we assumed these variables or parameters did not change before or after the data-available period." in row 195. Related references should be provided. Response: These assumptions were established based on available data and experience, and many other studies applied similar assumptions to develop the manure nutrient dataset. Related references have been added, meanwhile, the limitations and uncertainties of these assumptions were further discussed and explained in the discussion section 4.4. Assumption: "We assumed that spatial patterns of livestock distribution inside each county were static over the study period." The livestock distribution maps from the GLIMS dataset were the reference of the spatial pattern of manure nutrient production data within each county. The GLIMS data were developed by establishing statistical relationships between livestock inventory data and multiple environmental variables (e.g., climate, land cover, human activities), and using these relationships to predict livestock distributions across the globe. The GLIMS provided these standard maps of livestock distribution without considering the annual variation. We assumed the livestock distribution within each county was stable over the study period because the dynamic livestock maps were unavailable. A similar approach can be found in Potter et al. (2010). The environmental variables such as topography and landscape may be relatively stable over the study period, but some variables like human activities can change and induce the variation in livestock distribution inside each county. The accuracy of this manure nutrient production dataset would be improved if dynamic livestock maps are developed in the future. (L423-431) Assumption: "we assumed manure production changes in the U.S. were consistent with the global trend and manure N:P ratio was constant during 1860-1930" The manure nutrient production before 1930 was generated based on change rates in global manure N datasets from Holland et al. (2005). There is a period of overlap (1930-2004) between this global dataset and the USDA census data. During 1930-2004, the average annual change rates of manure N production were 1.08% in the global dataset and 1.01% in this study. Therefore, the changes in estimated manure N production in the U.S. before 1930 might be reasonable at a long-time scale. Similar assumptions can be found in Zhang et al. (2017). The ratio of N to P in animal manure varies among different animal species and changes along with proportions of different animal populations over time. From 1930 to 2017, the N:P ratio in the total manure production slightly decreased from 4.33 to 3.25. Due to the lack of manure P production data before 1930, we calculated the manure P production in this period according to manure N production and the constant N:P ratio in 1930. If the N:P ratio kept decreasing before 1930, the total manure P production may be overestimated during 1860-1929. (L432-443) Assumption: "we assumed these variables or parameters did not change before or after the data-available period." Changes in recoverability factors and crop yields over the study period were ignored due to the lack of data support and similar assumptions were made in Kellogg et al., (2000) and Puckett et al., (1998). With the development of livestock confinement facilities, the confinement and recoverability factors of animal manure may increase in recent decades (Kellogg et al., 2000). Hence, manure application can be overestimated before the 1980s and underestimated after the 2000s. The yields of different crops may change at different speeds over the study period, and that can affect the spatial patterns of manure nutrient demand of cropland as well as manure nutrient application. (L444-450)

Comment 5: Results: Statistical analysis should be described in more details in this section. Any detailed regional data to make comparison possible? Response: Thanks for pointing this out. We have added detailed numbers and P-values as well as more

analysis in the Results section. With regards to the comparison, we developed the annual crop nutrient demand maps during 1860-2017, and the comparison was based on the difference between manure nutrient production and crop nutrient demand maps. We used to put these nutrient maps in the supplementary and now we moved them (Fig 5) into the main text to make it clearer.

Comment 6: The first sentence of "3.1 temporal and spatial patterns of manure nutrient production"( in row 202), the author calculated the average total manure N and P production during 1860-2017. What's the meaning of this data in such a large time scale. Response: We used to use this average number to show the magnitude of manure N and P production over the study period, and we understand it may be not necessary. We have deleted this sentence and changed it to "We estimate that the total manure N and P production increased from 1.4 Tg N yr-1 and 0.3 Tg P yr-1 in 1860 to 7.4 Tg N yr-1 and 2.3 Tg P yr-1 in 2017, respectively.". (L227-228)

Comment 7: Similar to the row 234 (the average annual manure N and P application in the U.S.). In row 204, please rewrite the sentence "Compared with production rates in 1860 (1.4 Tg N yr-1 and 0.3 Tg P yr-1 204 ), manure N and P production in 2017 increased 5-fold and 7-fold at a rate of 0.04 Tg N yr-1 and 0.01 Tg P yr-1 205 , respectively". It's better to note that 1.4 Tg N yr-1 and 0.3 Tg P yr-1 is manure N and P production. Response: This sentence has been rephrased to "The total manure N and P production increased 5-fold and 7-fold during 1860-2017, with the increasing rates of 0.03 Tg N yr-2 and 0.006 Tg P yr-2 during 1860-1930, 0.05 Tg N yr-2 and 0.02 Tg P yr-2 during 1930-2017 (p<0.01), respectively." (L232-235)

Comment 8: The reason why the author chose the year of 1860, 1930, 1980, and 2017 for analysis should be specified in the manuscript. Response: The year of 1860 was the first year of the dataset and also the start period of industrialization. 1930 and 2017 were the earliest and latest years of the available USDA census data, respectively, and 1980 was chosen as the year at the middle of these two years. This content has been added as a note in the title of Figure 3. (L715-716)

Comment 9: In row 238, "total manure P application reached 2.3 Tg P yr-1 in 2017", please recheck the value of 2.3 Tg P yr-1 which does not match Fig.5. Response: We apologize for the mistake. The number is 0.6 Tg P yr-1, and it has been corrected in this sentence. (L273)

Comment 10: Discussion: In "4.1 Comparison with previous investigations", the authors have compared the results with four previous studies. So how to quantitatively describe the discrepancies caused by different coefficients or statistical methods? Response: Since these datasets used many different parameters and data sources and some of them were unavailable, it is difficult to quantify these differences caused by any specific coefficients or statistical methods. For example, in the USDA census report, data that would disclose operations of an individual farm are not published, and different studies use different approaches to deal with this issue, and the difference caused by these approaches cannot be directly quantified. But, we found an overlap period (1987-1997) that might allow us to compare our results with previous students. The average total manure N (P) production over 1987-1997 was 6.02 Tg N yr-1 (1.79 Tg P yr-1), 6.75 Tg N yr-1, 5.96 Tg N yr-1 (1.75 Tg P yr-1), 5.64 Tg N yr-1 (1.67 Tg P yr-1), and 6.01 Tg N yr-1 (1.86 Tg P yr-1), respectively, for this study, FAOSTAT, NuGIS, Kellogg et al., and Yang et al. The differences between different datasets were derived from calculation methods, chosen livestock types and numbers, as well as parameters, such as animal-specific excreted manure nutrient rates and the number of days in the life cycle. In terms of changing trends, manure N and P production were relatively stable after the 1960s in FAOSTAT, Kellogg et al., and Yang et al., while NuGIS data increased slightly between 1987 and 2007 and then decreased sharply after 2010. In contrast, our results showed an increasing trend after the 1980s due to the consideration of the increased animal body sizes. It is important to note that manure production can still increase even though the livestock number became stable. (L321-330)

Comment 11: Data availability: Since the author have cited the data resource in Methods, this part can be moved to Methods. Response: Data availability is a required

section in the journal ESSD. We cannot move it into Methods.

Comment 12: Using words/phrases: The language of this manuscript should be re-polished. Response: Done.

Reviewer 2 Comment 13: This article presents a dataset about manure N and P production and use in the United States of America and its evolution in time. It is globally well written. However, my opinion is that it lacks of precision, discussion about main hypothesis and clarification of the main findings. Response: We greatly appreciate the reviewer's thorough review of the paper and the valuable suggestions offered. We feel that this manuscript has been greatly improved by addressing these comments. The main hypotheses and main findings have been clarified in the manuscript. Our detailed responses and modifications to the text can be found in the point-by-point responses to the reviewer's comments below.

Comment 14: More particularly, in the introduction, nutrients in manure are mainly presented without considering nutrients in mineral fertilization while they are related with them by the biophysical N and P cycles including the environmental consequences of losses of some of N and P components. Response: We have rewritten the second paragraph in the Introduction according to your comment, and the nutrients in manure and fertilizer have been combined to introduce the N and P cycles. "Although the application of manure and fertilizer has enhanced crop production, excessive nutrient might leave the Soil-Plant-Animal system through the biogeochemical flow and potentially contaminate the environment if not properly managed (Mueller and Lassaletta, 2020; Zanon et al., 2019). Specifically, agricultural land is the sink for anthropogenic N and P inputs (e.g. synthetic fertilizer, manure, atmospheric deposition), and simultaneously acts as N and P sources for aquatic systems as well as a N source for the atmosphere (Bouwman et al., 2013; Elser and Bennett, 2011; Schlesinger and Bernhardt, 2013). The major N gaseous losses from fertilizer use and animal excreta include the emissions of ammonia ($NH_3$), nitrous oxide ($N_2O$), and nitric oxide. $NH_3$ can react with other air pollutants and form aerosols to reduce visibility and threaten human health

(Bouwman et al., 2002; Xu et al., 2018), and N2O is one of the most important green-house gasses (Davidson, 2009). N2O emission from animal manure is one of the major contributors to global anthropogenic N2O emissions (Tian et al., 2020). Additionally, large fractions of applied N and P in cropland lose through leaching, erosion, and surface runoff and then are transported into rivers toward lakes and coastal oceans (Smith et al., 1998; Van Drecht et al., 2005). Excess N and P could dramatically impair freshwater and coastal ecosystems, causing eutrophication, hypoxia, and fish-killing (Garnier et al., 2015; Smith et al., 2007)." . (L69-84)

Comment 15: In the introduction more precision is required, for example in definition and characterization of manure. (line 70 and followings). For example urea is not always the main N component and is not always readily usable by plant (toxicity for young plants). N composition of manure depend of many factors (excretion, : : : fermentation dynamic and thus evolution during manure storage). Response: To avoid confusion, we deleted this part of content in the Introduction section. The new content can be found in comment #14.

Comment 16: There is confusion between excretion and manure after storage. So it is not always clear what is included within the term manure or N produced (N straw included?, only excreted? Before or after volatilization?). Response: Thanks for pointing this out. The manure production in this study refers to the animal excretion (only excreted and before volatilization). We clarified it in the Method section 2.1. (L125)

Comment 17: I would also emphasize, in a circular approach, the notion of N and P source and sink at local/global level. Response: We added it in the Introduction. "Specifically, agricultural land is the sink for anthropogenic N and P inputs (e.g. synthetic fertilizer, manure, atmospheric deposition), and simultaneously acts as N and P sources for aquatic systems as well as a N source for atmosphere". (L72-73)

Comment 18: There is a lack of clear identification of the mean objectives of the paper and the underlying hypotheses. Response: We have rewritten the third paragraph to

make the objectives clear. The hypotheses were further discussed in the Discussion section 4.4. The third paragraph has been rewritten as: "Considering the importance of manure nutrient on crop production, greenhouse gas emission, and water pollution, it is vital to have a better understanding of livestock manure production and application at national or even global scales (Potter et al., 2010; Sheldrick et al., 2002; Tian et al., 2016). Precise and spatialized manure nutrient data can benefit stakeholders from finding available local recyclable nutrient sources or making strategies to minimize N and P losses. Currently, most studies only provided county-level manure nutrient production data in the U.S., with relatively short periods (Kellogg et al., 2000; Ruddy et al., 2006). Nevertheless, terrestrial biosphere models usually require spatially explicit manure nutrient input data to simulate the anthropogenic effect on biogeochemical cycles since the preindustrial period (Tian et al., 2019). Studies focusing on soil nutrient storage change and legacy soil nutrient also need long-time series manure nutrient data (MacDonald et al., 2012; Rowe et al., 2016). Moreover, previous studies usually assumed that nutrient excretion per animal is constant over time when quantifying nutrient production based on livestock number, which may lead to uncertainties (Zhang et al., 2020). Geographically explicit manure nutrient application in cropland (excluding pasture), as the direct nutrient input for the soil-crop system, hasn't been specifically estimated across the U.S. In this study, our objectives are to (1) develop grid-level manure N and P production datasets in the U.S. based on county-level livestock populations, dynamic livestock weight over time, and high-resolution livestock distribution maps; (2) develop grid-level manure N and P application in cropland datasets by integrating manure nutrient production and nutrient demand of crops; (3) investigate the spatiotemporal patterns of manure nutrient production and application based on these datasets, and (4) further identify regions with a high risk of excessive nutrient loading. The four datasets display the masses of manure N and P per area in each $30\times30$ arc-second grid-cell during 1860-2017. The datasets can be used to drive ecosystem, land surface, and hydrological models to simulate manure-induced greenhouse gas emissions and nutrient loadings.". (L89-112)

Comment 19: In the method section I would add a summary explaining the main steps as introduction of the method section. Response: Thanks for this suggestion. We have added a summary at the beginning of the method section. "Datasets of manure N and P production and application were developed by incorporating multiple datasets (Table 1). The geographically explicit manure N and P production data were first calculated based on county-level livestock populations, dynamic livestock weights, and livestock distribution maps. Then the crop nutrient demand maps were developed by merging cropland distribution maps with crop-specific harvest area and nutrient assimilative capacities. Finally, the spatially explicit manure N and P application data were estimated by incorporating county-level manure production, recoverability factors, cropland fraction, and cropland nutrient demand maps." (L114-121)

Comment 20: When information from other works or methods are used, like at line 120, i recommend to explain them briefly and add some summarizing values ( as for Line 166 manure nutrient recoverability rate ) Response: According to your suggestion we added corresponding information in the method part. "Livestock population data for the recent five census reports (1997–2017) can be directly collected from the USDA Census Data Query Tool. Livestock population data before 1997 were collected from Cornell Institute for Social and Economic Research Data Archive (1949–1992), or manually digitalized from the USDA reports (1930–1945). More details of data collection, methods of dealing with missing data can be found in Yang et al. (2016)." (L139-143) "$Rf_{x,c}$ is the manure nutrient recoverability rate (the recoverability rates are unitless and county-specific with average values 0.19 for N and 0.35 for P)". (L191-192)

Comment 21: Some hypothesis are done and should be better explained eg. Line 126 (spatial distribution of livestock). Response: The livestock distribution maps from the GLIMS dataset were the reference of the spatial pattern of manure nutrient production data within each county. The GLIMS data were developed by establishing statistical relationships between livestock inventory data and multiple environmental variables (e.g., climate, land cover, human activities), and using these relationships to predict

livestock distributions across the globe. The GLIMS provided these standard maps of livestock distribution without considering the annual variation. We assumed the livestock distribution within each county was stable over the study period because the dynamic livestock maps were unavailable. A similar approach can be found in Potter et al. (2010). The environmental variables such as topography and landscape may be relatively stable over the study period, but some variables like human activities can change and induce the variation in livestock distribution inside each county. The accuracy of this manure nutrient production dataset can be improved when dynamic livestock maps are developed in the future. This hypothesis has been further explained in the discussion section 4.4. (L423-431)

Comment 22: I recommend to use only SI metrics. Response: All units have been changed to SI metrics.

Comment 23: Regarding the statistics presented I wonder if it would not be more appropriate to present them as median and percentiles. Response: We want to show the inter-annual variation of manure production, and we understand the average value may be not proper here. We have removed the average value and used the specific value of each year instead. Median and percentile could not present information regarding the temporal variation of manure nutrient production. (L227-228)

Comment 24: In the result and the discussion sections there is a need for better explanation of the results observed at local scale and their consequences. Response: Thanks for this suggestion. We have added more detail and explanation of the results in the results section.

Comment 25: I would also expect discussion about the relation between P and N evolution if possible including mineral fertilization or biological fixation of nitrogen. Response: We agree with your suggestion that it is interesting to investigate the relation between P and N evolution. In our study, the N:P ratio in the total manure production changed from 4.33 in 1930 to 3.25 in 2017, meanwhile, the N:P ratio in the total manure

application decreased from 2.62 to 2.32 during 1930-2017. Excretion of milk cows and horses have relatively high N:P ratios (5.6-6.6), while excretion of beef cows and broilers have relatively low N:P ratio (3-3.2). The decrease in N:P ratio in the total manure production was related to the change in the structure of animal population, specifically, the proportion of beef cows and broilers increased while that of milk cows decreased over the study period. This part of content has been added in the results section 3.1. (L235-239, L273-277) To investigate the N and P evolution in agricultural systems, it is better to include mineral fertilization and biologically nitrogen fixation (BNF). We have searched literature and collect datasets of fertilizer and BNF. Currently, most studies only provide county-level nutrient dataset after the 1980s. We have compared the manure data in this study with the dataset from the NuGIS (Tables 1 and 2). It showed that manure N and P application accounted for relatively small shares compared with synthetic fertilizer. N and P balances demonstrated that N kept a surplus state and P fluctuated between surplus and deficit during 1987-2014. If we want to conduct this analysis at a long-time scale, we need to collect or develop fertilizer, N fixation, and nutrient in harvested crop datasets with the same spatial resolution and long-time series. Therefore, estimating nutrient status may involve huge amounts of work. For the current study, we may focus on the manure nutrient, and we will definitely conduct the study of N and P budget and evolution in the U.S. in our future work. We believe more meaningful conclusions can be found based on a more robust independent study.

Comment 26: I was also surprised by the difference for P between production and application. The fate of P and the losses have to be explained and validated. Response: Manure can be deposited on cropland by grazing animals, but is commonly transported from animal confinement and manure storage facilities and spread on the ground or injected into it. However, opportunities for widespread manure substitution are limited: manure can be costly to transport for even short distances, and some crops are far from sources of manure production. Moreover, manure may not have the precise combination of nutrients needed for specific crops and fields. The county-level recoverability factors in this study were provided by NuGIS, and the average manure

N recoverability in the entire U.S. was 0.19 while average manure P recoverability was 0.35. These recoverability factors were calculated based on the USDA report (Kellogg et al., 2000). The low values of recoverability were because the recoverability factors of beef, cows, calves, heifers, and stockers were very low (around 0.05-0.2) and these animals contributed a large amount of manure production. Additionally, the recoverable manure can be applied to both cropland and pastureland, and we subtracted the amount of manure that was potentially applied on pastureland in this study. All these reasons lead to the low manure application and we have discussed it in the discussion section 4.1. (L351-353)

Comment 27: In the discussion section the main hypotheses like those of line 126 cattle distribution ot line 176 regarding evolution of crop yield over time have to be discussed. Response: Thanks for your suggestions. We have added discussion regarding these two hypotheses in section 4.4. The first one has been explained under comment #21. Changes in recoverability factors and crop yields over the study period were ignored due to the lack of data support and that may cause a bias in quantifying manure nutrient application. With the development of livestock confinement facilities, the confinement and recoverability factors of animal manure may increase in recent decades (Kellogg et al., 2000). Hence, manure application can be overestimated before the 1980s and underestimated after the 2000s. The yields of different crops may change at different speeds over the study period, and that can affect the spatial patterns of manure nutrient demand of cropland as well as manure nutrient application. (L444-450)

Comment 28: Can the peak and decrease in N production around 2008 be explained? Response: Yes. The slight decrease in manure nutrient production between 2008 and 2012 may be associated with the financial crisis and the low demand for livestock products. It has been added in results section 3.1. (L231-232)

Comment 29: I would add data in the section lines 280-285 and expect explanation about the observed differences. Response: The number and explanation have been added. "The average total manure N (P) production over 1987-1997 was 6.02 Tg N

yr-1 (1.79 Tg P yr-1), 6.75 Tg N yr-1, 5.96 Tg N yr-1 (1.75 Tg P yr-1), 5.64 Tg N yr-1 (1.67 Tg P yr-1), and 6.01 Tg N yr-1 (1.86 Tg P yr-1), respectively, for this study, FAOSTAT, NuGIS, Kellogg et al., and Yang et al. The differences between different datasets were derived from calculation methods, chosen livestock types and numbers, as well as parameters, such as animal-specific excreted manure nutrient rates and the number of days in the life cycle.". (L321-327)

Comment 30: To me the conclusion is more a summary than a conclusion. It can also be improved. Response: Thanks for pointing this out. We have revised the Conclusion section and make it clearer to present our key findings. "Manure nutrient production and application in the livestock-crop system substantially altered the regional and global N and P cycle. In this study, we developed geographically explicit datasets of animal manure N and P production and their application in cropland across the contiguous U.S. from 1860 to 2017. The dataset indicated that both manure N and P production and application significantly increased over the study period. Although livestock numbers became stable in recent decades, manure nutrient production still increased due to the enhanced livestock body weight after the 1980s. Enhanced livestock weights contributed 59% and 54% of the increase in manure N and P production, respectively, from 1987 to 2017. Meanwhile, manure nutrient production intensified and enlarged inside the Midwest and toward the Southern U.S. from 1980 to 2017, and became more concentrated in numerous hot spots. As manure nutrient application also expanded toward the Southeastern coastline, massive amounts of nutrients get more of a chance to be transported to the estuary. The enrichment of manure nutrient in the South Atlantic-Gulf, Mid-Atlantic, and Mississippi River basins increased the risk of excessive nutrient loading into the Gulf of Mexico and the Atlantic Ocean under extreme weather conditions (e.g., hurricane). Therefore, it is of great importance to effectively store, utilize, and transport animal manure in order to reduce nutrient pollution and restore the environment." (L467-482)

Comment 31: References have to be added for all the statements like at line 53,

69 I would reformulate lines 54 to 60, 66 to 69, 211. Response: All the references have been added and these sentences have been reformulated. "The circular nutrient source provided by manure enables nations to sustain their agricultural production with less reliance on imported fertilizer, especially mineral P fertilizer (Koppelaar and Weikard, 2013; Powers et al., 2019). Different from N which can be fixed from the atmosphere through microbial symbiosis with plants and the Haber-Bosch process, P is a rock-derived nutrient and there is no biological or atmospheric source for P." (L58-62) "Although the application of manure and fertilizer has enhanced crop production, excessive nutrient might leave the Soil-Plant-Animal system through the biogeochemical flow and potentially contaminate the environment if not properly managed (Mueller and Lassaletta, 2020; Zanon et al., 2019). Specifically, agricultural land is the sink for anthropogenic N and P inputs (e.g. synthetic fertilizer, manure, atmospheric deposition), and simultaneously acts as N and P sources for aquatic systems as well as a N source for atmosphere (Bouwman et al., 2013; Elser and Bennett, 2011; Schlesinger and Bernhardt, 2013)." (L69-75) "The distribution maps showed that the Midwestern U.S. (e.g., Iowa, Missouri, and Illinois) was the core region (> 300 kg N km-2 yr-1 or 100 kg P km-2 yr-1) of manure N (P) production in 1860. From 1860 to 1930, the high manure nutrient production region (> 600 kg N km-2 yr-1 or 200 kg P km-2 yr-1) mainly enlarged outwards from the Midwest. Between 1930 and 1980, the manure N (P) production not only intensified in the Midwest but also in the Southern U.S. (e.g., Texas, Georgia, and North Carolina). After 1980, manure N (P) production became more concentrated in many hot spots (> 6000 kg N km-2 yr-1 or 3000 kg P km-2 yr-1), especially in the southeastern U.S." (L241-248)

Comment 32: Line 174 maize instead of Maize Line 179 - =1? Line 203 (mean standard deviation) in the material and method section Lines 254 -256 in the material and method section Response: The "maize" has been revised. (L200) The resolution of the data was 1 km×1 km. (L205) This sentence with the mean value has been removed. This sentence has been moved to the summary part of the method section. (L121-123)

Comment 33: Line 274 remove the "our" Line 282 Kellogg et al. (2000), and Yang et al. (2016) within Response: The word has been removed. "Kellogg et al. and Yang et al." has been used instead. (L328)

Please also note the supplement to this comment:
https://essd.copernicus.org/preprints/essd-2020-185/essd-2020-185-AC1-supplement.pdf

[Figure]

**Fig. 1.** Figure 1 Comparing total production, application, and demand of manure N and P in the contiguous U.S. from 1860 to 2017

Table 1 The total N budget in the U.S. (Tg N/yr)

| Year | N fertilizer | manure production | manure application | N fixation by legumes | Removed N by harvested crop | N Balance |
|---|---|---|---|---|---|---|
| 1987 | 9.89 | 5.67 | 0.88 | 5.02 | 11.82 | 3.98 |
| 1992 | 11.01 | 5.87 | 0.94 | 5.38 | 12.86 | 4.46 |
| 1997 | 11.85 | 6.60 | 1.09 | 6.32 | 14.61 | 4.66 |
| 2002 | 11.04 | 6.76 | 1.15 | 6.44 | 14.60 | 4.04 |
| 2007 | 12.59 | 7.43 | 1.28 | 6.64 | 15.85 | 4.67 |
| 2010 | 11.86 | 7.17 | 1.24 | 7.22 | 17.14 | 3.18 |
| 2012 | 13.21 | 7.00 | 1.22 | 6.38 | 15.47 | 5.33 |
| 2014 | 12.96 | 7.16 | 1.25 | 7.50 | 18.25 | 3.47 |

**Fig. 2.** Table 1 The total N budget in the U.S. (Tg N/yr)

Table 2 The total P budget in the U.S. (Tg P/yr)

| Year | P fertilizer | Manure P production | Manure P application | Removed P by harvested crop | P balance |
|---|---|---|---|---|---|
| 1987 | 1.68 | 1.68 | 0.39 | 1.77 | 0.30 |
| 1992 | 1.76 | 1.76 | 0.42 | 1.94 | 0.24 |
| 1997 | 1.93 | 1.94 | 0.48 | 2.19 | 0.21 |
| 2002 | 1.91 | 1.96 | 0.49 | 2.17 | 0.23 |
| 2007 | 1.89 | 2.15 | 0.54 | 2.39 | 0.04 |
| 2010 | 1.74 | 2.11 | 0.52 | 2.57 | -0.30 |
| 2012 | 1.85 | 2.08 | 0.52 | 2.32 | 0.06 |
| 2014 | 2.02 | 2.16 | 0.54 | 2.74 | -0.19 |

**Fig. 3.** Table 2 The total P budget in the U.S. (Tg P/yr)